Review

Subject Area:
biochemistry/cellular biology/molecular biology

Keywords:
SUMOylation, cholesterol homeostasis, ubiquitin-like

Authors for correspondence:
Ana Talamillo
e-mail: atalamillo@cicbiogune.es
Rosa Barrio
e-mail: rbarrio@cicbiogune.es

# SUMOylation in the control of cholesterol homeostasis

Ana Talamillo[1], Leiore Ajuria[1], Marco Grillo[2,3], Orhi Barroso-Gomila[1], Ugo Mayor[4,5] and Rosa Barrio[1]

[1]Center for Cooperative Research in Biosciences (CIC bioGUNE), Basque Research and Technology Alliance (BRTA), Bizkaia Technology Park, Building 801A, 48160 Derio, Spain
[2]Institut de Génomique Fonctionnelle de Lyon (IGFL), École Normale Supérieure de Lyon, Lyon, France
[3]Centre National de la Recherche Scientifique (CNRS), Paris, France
[4]Department of Biochemistry and Molecular Biology, Faculty of Science and Technology, University of the Basque Country (UPV/EHU), Leioa, Spain
[5]Ikerbasque, Basque Foundation for Science, Bilbao, Bizkaia, Spain

MG, 0000-0003-2155-0645; RB, 0000-0002-9663-0669

SUMOylation—protein modification by the small ubiquitin-related modifier (SUMO)—affects several cellular processes by modulating the activity, stability, interactions or subcellular localization of a variety of substrates. SUMO modification is involved in most cellular processes required for the maintenance of metabolic homeostasis. Cholesterol is one of the main lipids required to preserve the correct cellular function, contributing to the composition of the plasma membrane and participating in transmembrane receptor signalling. Besides these functions, cholesterol is required for the synthesis of steroid hormones, bile acids, oxysterols and vitamin D. Cholesterol levels need to be tightly regulated: in excess, it is toxic to the cell, and the disruption of its homeostasis is associated with various disorders like atherosclerosis and cardiovascular diseases. This review focuses on the role of SUMO in the regulation of proteins involved in the metabolism of cholesterol.

## 1. Introduction

Cholesterol plays essential roles in the cellular organization and takes part in a variety of intracellular mechanisms. The disruption of cholesterol homeostasis associated with several diseases reveals its importance in human health. For example, defects in cholesterol biosynthesis cause the Smith–Lemli–Opitz syndrome, and low cholesterol levels are associated with the risk of neuropsychiatric disorders [1]. By contrast, excessive cholesterol in the body is associated with cardiovascular diseases, and several studies show that increased serum cholesterol levels are correlated with the risk of developing cancer and with cancer progression (reviewed by [2]). Ubiquitylation leading to protein degradation has been linked to mammalian cholesterol homeostasis [3]. However, there is limited information regarding other post-translational modifications by members of the ubiquitin-like family of proteins (UbLs). Here, we focused on the role that modifications by the small ubiquitin-related modifier (SUMO) exert in proteins involved in cholesterol biosynthesis, uptake, transport and secretion, that is in the regulation of cholesterol homeostasis.

## 1.1. SUMO family members

SUMO covalently attaches to target proteins in a process termed SUMOylation. SUMO proteins are present in all eukaryotes and are highly conserved across species. Only one *SUMO* gene has been identified in the yeast *Saccharomyces cerevisiae*, the nematode *Caenorhabditis elegans* and the insect *Drosophila melanogaster*, whereas four *SUMO* paralogues are found in mammals and at least eight in

royalsocietypublishing.org/journal/rsob   Open Biol. **10**: 200054

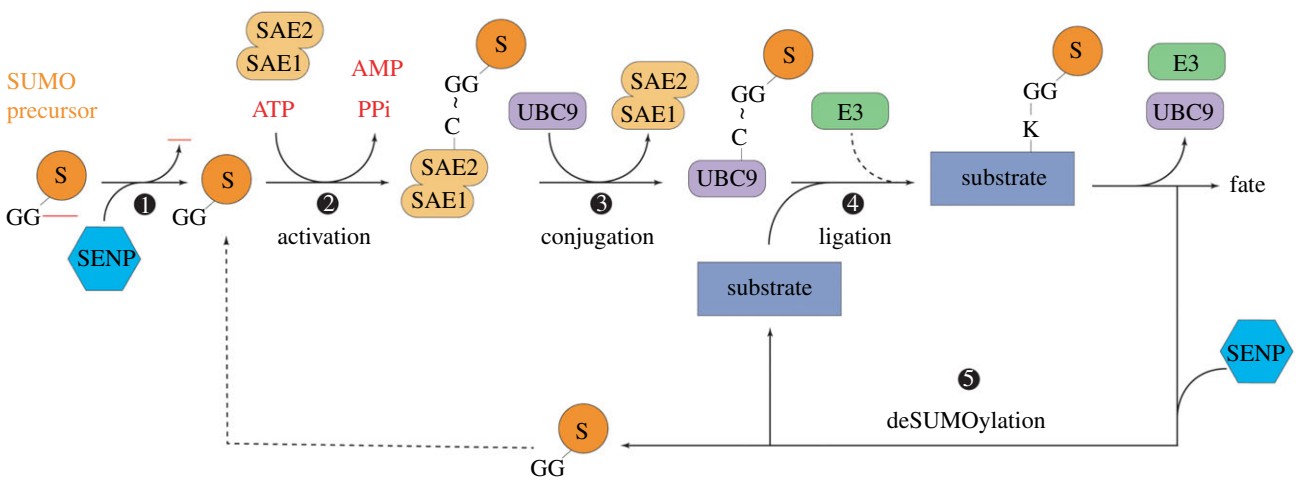

**Figure 1.** The SUMOylation cycle. SUMO precursor is the first processed for activation by SENPs (1). Mature SUMO forms a thioester bond with the heterodimer AOS1/UBA2-activating enzyme in an ATP-dependent manner (2: activation). E1–SUMO then passes SUMO to UBC9, which also forms a thioester bond (3: conjugation). E2–SUMO can directly modify substrates, but the action of E3s enhances conjugation rates by binding either E2–SUMO or substrates (4: ligation). SUMO and the substrate can be recycled by the action of an SENP (5: deSUMOylation). S: SUMO; ∼: thioester bond.

plants (revised by [4]). The human genome contains four SUMO genes (*SUMO1–4*). SUMO2 and SUMO3 share 97% identity in their amino acid sequence and are referred to as the SUMO2/3 subfamily. SUMO1 shares only 47% sequence identity with SUMO2/3. Finally, SUMO4 shares 87% identity with SUMO2, but it has only been detected in few tissues [5]. Several evidences indicate important differences between these *SUMO* paralogues: SUMO1 and SUMO2/3 differ in their cellular distribution and usually conjugate to different substrates. In addition, the rate of conjugation/deconjugation is higher for SUMO2/3 than for SUMO1 [6]. There are also differences in their ability to form polySUMO chains. While SUMO2/3 can be itself SUMOylated in the N-terminal part forming chains, SUMO1 is considered as a stopper of SUMO chain formation, because it lacks the lysine targeted for SUMO conjugation [7]. Unlike vertebrate SUMO2/3, *Drosophila* SUMO does not form polySUMO chains [8]. By contrast, *C. elegans* SMO1 is involved in both mono-SUMOylation and SUMO chain formation [9].

## 1.2. The SUMOylation pathway

Similarly to the modifications by other members of the UbL family, SUMOylation results in the formation of an isopeptide bond between a carboxyl group of SUMO and a lysine residue in the target proteins [5,10–12]. The SUMOylation pathway consists of several steps, each carried out by one or more specific enzymes: SUMO processing, activation, conjugation and ligation. As SUMOylation is a reversible process, deSU-MOylation relies on particular proteases, resulting in a very dynamic process (figure 1).

SUMO proteins are synthesized as immature polypeptide precursors that need to be processed for activation. Maturation happens through proteolysis of the C-terminal end, revealing a di-glycine motif essential for its further conjugation to target proteins. The proteases involved in such maturation are the ubiquitin-like protein-specific proteases (Ulps) in yeast and invertebrates (Ulp1 and Ulp2 in *S. cerevisiae* and *D. melanogaster*; ULP-1, -2, -4 and -5 in *C. elegans*), and sentrin-specific proteases (SENPs) in mammals (SENP1–3 and SENP5–7) [13]. Both Ulps

and SENPs are also involved in the deconjugation of SUMO from substrates.

Processed SUMO is activated by the formation of a thioester bond with the activating E1 enzyme. The E1 enzyme is a heterodimer composed of SUMO-activating enzyme subunit 1 (SAE1, or Aos1) and SUMO-activating enzyme subunit 2 (SAE2, or Uba2) [14,15]. This heterodimer activates the C-terminus part of SUMO in a two-step reaction. First, the SAE1 subunit adenylates SUMO using ATP. In a second step, the adenylated SUMO is attacked by the catalytic cysteine of the E1, forming the E1–SUMO thioester bond. E1 then transfers the SUMO load to the conjugating E2 enzyme by transthioesterification (figure 1).

There is a single SUMO E2-conjugating enzyme, UBC9, which is highly conserved in eukaryotes. E2s receive the activated SUMO in the ubiquitin-conjugating catalytic (UBC) fold, where the catalytic cysteine is located, to form an E2–SUMO thioester bond interaction. Once conjugated, E2s can transfer SUMO to a lysine residue in the target protein, either directly or by a handful of E3 ligases (figure 1). Although the SUMO–E2 can often bind directly to substrates and transfer them the SUMO moiety, E3 protein ligases increase the rate of conjugation through two mechanisms: either binding E2–SUMO thioester and catalysing the transthioesterification between E2 and the substrate, or directly binding the substrate and recruit E2–SUMO to facilitate SUMO conjugation to the substrate [12].

SUMO conjugation is a very dynamic, reversible and tightly controlled process. As previously mentioned, Ulp and SENP proteases fulfil a dual role by processing and deconjugating SUMO from substrates [13]. Their activity relies on the conserved C-terminus catalytic domain that induces a conformational change between SUMO and the substrate to facilitate the hydrolysis of the covalent bond [16–19].

In addition to the covalent attachment of SUMO to lysine residues in target proteins, the SUMO-interacting motifs (SIMs) mediate non-covalent interactions with SUMO or SUMO-conjugated proteins (reviewed in [20]). The functional consequences of SUMO binding to SIM-containing proteins include the recruitment of proteins to subcellular localizations, the formation of promyelocytic leukaemia

protein (PML) nuclear bodies or the recruitment of repressors complexes to the chromatin [21,22].

## 1.3. Role of SUMOylation in the control of gene expression

SUMOylation affects the function of proteins involved in many different metabolic pathways including those implicated in cholesterol homeostasis. In many instances, this is achieved through the regulation of transcription factors and chromatin regulators.

The conjugation of SUMO to transcription factors correlates usually with repressed transcription, but there are also examples of proteins whose activity is enhanced by this modification [23]. For example, the SUMO acceptor sites in several transcription factors, such as the glucocorticoid receptor, the mineralocorticoid receptor and SERBP-1, map to known inhibitory motifs [24]. SUMOylation can modify the transcriptional output by changing different aspects necessary for the activity. For instance, SUMO modification can change the subnuclear localization or the nuclear import/export of certain factors [25]. SUMO can also alter the DNA-binding capacity of other factors, as shown, for example, with SUMO1 modification of the heat shock transcription factor 2 (HSF2) [26]. SUMOylation interferes as well with other post-translational modifications such as acetylation, phosphorylation and ubiquitination [27]. The competition between acetylation and SUMOylation usually represents a switch between active and inactive forms of the transcription factor [28]. SUMOylation can also influence the phosphorylation even if the modifications occur on different amino acid residues such as in the transcription factor STAT5 [29]. Similarly, by inhibiting ubiquitination, SUMO can control the stability of the transcription factors [30]. A link between SUMO and ubiquitin pathways in the control of genome stability involves the SUMO-targeted ubiquitin ligases (STUbLs) which bind via SIM motifs to SUMOylated proteins and target them for ubiquitylation [31].

In addition to regulate the specific action of transcription factors, SUMOylation has an important role in chromatin remodelling by modifying histones and other chromatin-associated proteins [32,33]. All core histones can be SUMOylated. In the case of H4, SUMOylation increases its interaction with histone deacetylase HDAC1 and the heterochromatin protein 1 HP1γ, which is in agreement with a role for SUMO in transcriptional repression [34]. More recently, SUMO has been shown to be necessary for deposition of the H3K9me3 molecular hallmark of heterochromatin associated with gene silencing [35,36]. Furthermore, SUMO is required for piRNA-guided deposition of repressive chromatin marks necessary for transcriptional silencing [37]. Therefore, SUMO is crucial in gene expression regulation at multiple levels by modifying the properties of specific transcription factors and by regulating the chromatin structure.

## 1.4. Cholesterol homeostasis

Cholesterol is an essential component of the cell, which is involved in the permeability and fluidity of the cell membrane and in the modulation of transmembrane signalling pathways. It is also the precursor of all steroid hormones, vitamin D, oxysterols and bile acids, which regulate diverse metabolic pathways [38]. Although it plays these vital roles, high levels of intracellular cholesterol are toxic to the cells and its accumulation can lead to cardiovascular diseases. Therefore, the balance between cholesterol synthesis, absorption and excretion needs to be tightly regulated. Under low cellular cholesterol concentrations, the cell upregulates cholesterol intake and increases cholesterol synthesis. Under high cholesterol concentrations, oxysterols regulate cholesterol homeostasis by binding to the nuclear receptor liver X receptor (LXR) and increasing the removal of cholesterol as bile acids.

In vertebrates, the cholesterol sources are *de novo* synthesis and the dietary uptake. By contrast, arthropods and nematodes, which are unable to synthesized *de novo* sterols, must obtain cholesterol directly from the diet [39]. In mammals, the main cells involved in the synthesis of cholesterol are hepatocytes and enterocytes. Cholesterol is synthetized *de novo* from acetyl-CoA in the endoplasmic reticulum (ER) and the rate-limiting enzyme is the hydroxymethylglutaryl-CoA reductase (HMG-CoAR), which catalyses the synthesis of mevalonate. The cholesterol absorbed by enterocytes, a process regulated by the Niemann-Pick-C1-like-1 (NPC1L1) protein, is esterified with a fatty acid in the membrane of the ER [40,41]. The cholesteryl ester is further processed in the Golgi to form chylomicrons that are secreted to the circulation. Free cholesterol returns to the intestinal lumen via the transporters ATP-Binding Cassette Subfamily G Members 5 and 8 (ABCG5 and 8).

Cholesterol biosynthesis and absorption are both regulated by a sensor mechanism of cholesterol levels in the ER, which acts through the transcription factor family sterol regulatory element-binding proteins (SREBPs) [42]. SREBPs are translated as inactive precursors and retained in the ER membrane via association with SCAP (SREBP cleavage-activating protein). This protein contains a sterol-sensing domain: in the presence of sterol, SCAP–SREBP interacts with INSIG1 (insulin-induced protein 1) to prevent SREBP modification [43,44]. Under low cholesterol conditions, SCAP–SREBP dissociates from INSIG1 and moves to the Golgi where it is proteolytically cleaved releasing a mature nuclear protein. Once translocated to the nucleus, nSREBPs activate the transcription of genes involved in cholesterol synthesis and uptake.

The transport of cholesterol and other lipids through the body requires lipoproteins, which consist of a hydrophobic core containing cholesteryl ester and triacylglycerol, and a hydrophilic coat formed by phospholipids, free cholesterol and apolipoproteins. The major cholesterol-carrying lipoproteins in the blood are the low-density lipoprotein (LDL) and the high-density lipoprotein (HDL). The LDL particles transport cholesterol to cells that required lipids, while the HDL transfers excess of cholesterol from peripheral tissues to the liver. To prevent the toxic effect of cholesterol accumulation, several members of the family of nuclear receptors control the storage, transport and catabolism of sterols. For example, LXR prevents cholesterol accumulation in the enterocytes. In mammals, the synthesis and excretion of bile acids comprise the major cholesterol catabolism pathway, and the liver receptor homologue 1 (LRH-1/NR5A2), LXR and Farnesoid X receptor (FXR/NR1H4) are important regulators of these pathways (reviewed by [45]). The levels of bile acids are tightly regulated through the transcriptional regulation of the cytochrome P450 7A1 (CYP7A1), a rate-limiting enzyme in the pathway of bile acid biosynthesis [46]. As examples, LXRα upregulates *CYP7A1* mRNA and eliminates the excess of cholesterol via bile acid synthesis and excretion [47], whereas

FXR acts as a receptor for bile acids and inhibits bile acid synthesis by repressing CYP7A1 transcription [48].

## 2. SUMOylation in *de novo* cholesterol synthesis, uptake and storage

In the next sections, we provide information about the roles of SUMO modification in the maintenance of whole cell and body cholesterol balance. Specifically, we first discuss the role of SUMO in cholesterol synthesis and uptake, and SUMOylation of the SREBP as a master regulator of these processes.

### 2.1. SUMOylation in the mevalonate pathway

The cholesterol biosynthetic pathway, also referred to as the mevalonate pathway, plays a critical role in cholesterol homeostasis. This pathway promotes, through a series of enzymatic steps, the conversion of acetyl-CoA first into mevalonate and, eventually, into farnesyl diphosphate, the main precursor of sterol isoprenoids such as cholesterol, steroid hormones and bile acids, as well as of non-sterol isoprenoids. Despite the fundamental metabolic role of this pathway, little is known about the post-translational mechanisms regulating the activity of the enzymes involved.

The first committed step of the mevalonate pathway is catalysed by the enzyme HMG-CoA synthase (HMGS-1). The mevalonate pathway in *C. elegans*, although it lacks the cholesterol synthesis branch, is very well conserved. In *C. elegans*, HMGS-1 function is negatively regulated and inactivated by SUMO. Sapir *et al.* [49] found HMGS-1 to be SUMOylated *in vivo* and showed that knocking-down *smo1* by RNAi abolishes almost completely its SUMOylation.

In *C. elegans*, SUMOylation is physiologically balanced by the deSUMOylation activity of the protease ULP-4. Interestingly, ULP-4 appears to be regulated in an age-dependent manner. As the worm ages, ULP-4 undergoes cytoplasm-to-mitochondrial sequestration. The reduced cytoplasmic levels of ULP4 fail to balance the SUMOylation activity of smo1, resulting in increased SUMOylation and inactivation of HMGS-1. This substantially impairs the pathway, producing serious metabolic defects in the ageing worm. These defects can be partially rescued by supplying mevalonate with the diet, suggesting that this age-related metabolic effect depends mostly—if not only—on HMGS-1 activity. The SUMOylation of the human HMGCS1 *in vitro* and the conservation of the mevalonate pathway suggest that this mechanism may play similar roles in humans. However, the physiological relevance of SUMO in regulating the mevalonate pathway in other organisms outside nematodes requires further studies.

### 2.2. SUMO function in sterol uptake for steroidogenesis

Cholesterol is the most abundant sterol in insects even though, as mentioned, insects are not able to synthesize cholesterol *de novo*. Despite this difference when compared with vertebrates, *D. melanogaster* has been an excellent model for studying the molecular mechanisms that regulate cholesterol homeostasis. As an example, the vertebrate LXRs and related *Drosophila* DHR96 play similar regulatory roles in the control of cholesterol metabolism [50,51].

The *Drosophila* SUMO homologue, Smt3, is required for cholesterol uptake in steroidogenic tissues during post-embryonic development [52,53]. The sterol intake in these tissues is particularly important because cholesterol is the precursor for the steroid hormones, ecdysone and its derivative 20-hydroxyecdysone. These hormones control several aspects of the physiology including larval molting and metamorphosis. This function of SUMO in steroidogenic tissues is partially dependent on the nuclear receptor Fushi tarazu transcription factor 1, Ftz-f1. SUMO is required for the expression of Ftz-f1, which is also modified by SUMO *in vitro* and *in vivo*, leading to a reduced transcriptional activity. The SUMOylation of Ftz-f1 affects the expression of the scavenger receptor Snmp1 [54], which is required for cellular cholesterol uptake in the steroidogenic tissues and subsequent for steroid synthesis [52]. The mammalian Ftz-f1 subfamily of nuclear receptors, LRH-1 and steroidogenic factor 1 (SF1/NR5A1), are as well modified by SUMO [55–57] and play a central role in the control of cholesterol homeostasis (see §4). In addition, SUMO and ubiquitin regulate in mammals the function of other nuclear receptors in steroidogenesis [58].

### 2.3. SUMOylation of SREBP-2 inhibits its transcriptional activity

SREBP are transcription factors of the basic helix-loop-helix leucine zipper (bHLH-Zip) family, which comprises three isoforms in mammals: SREBP-1a, SREBP-1c and SREBP-2. SREBP-2 is the main isoform controlling the expression of genes involved in cholesterol metabolism; it regulates the expression of genes in the cholesterol biosynthetic pathway and genes encoding proteins required for cholesterol uptake [59,60]. When the cellular sterol concentrations decrease, SREBPs translocate to the nucleus and upregulate the expression of these genes. In invertebrates such as *C. elegans* and *Drosophila*, there is only one SREBP, which resembles the vertebrate SREBP-1c and controls fatty acid biosynthesis [61,62]. In these organisms, the *Drosophila* dSREBP and the *C. elegans* SBP-1 do not seem to be involved in cholesterol homeostasis; their activities are regulated by phospholipids, but not by sterols [63,64].

SREBP regulation in the nucleus is driven by post-translational modifications such as phosphorylation and SUMOylation. SREBP-2, which interacts with the E2 SUMO-conjugating enzyme UBC9, is modified by SUMO1 at a single site (Lys464) [65]. This modification represses its transcriptional activity, while phosphorylation of SREBP-2 at S455, in close proximity to the SUMOylation site, increases its transcriptional activity [66–68]. Mutational analysis showed that phosphorylation and SUMOylation act as mutually exclusive competitive antagonistic signals (figure 2). SREBP SUMOylation inhibits transcription indirectly through the recruitment of a co-repressor complex that includes histone deacetylase 3 (HDAC3). Although no direct interaction was observed between SUMOylated SREBP-2 and HDAC3, the presence of HDAC3 is necessary for the inhibition of the transcriptional activity [68].

## 3. Cholesterol-mediated LXR SUMOylation regulates inflammation

High levels of cholesterol induce the transcriptional activity of the nuclear receptor LXRs, which form obligate heterodimers with retinoid X receptors (RXRs). Oxysterols, which

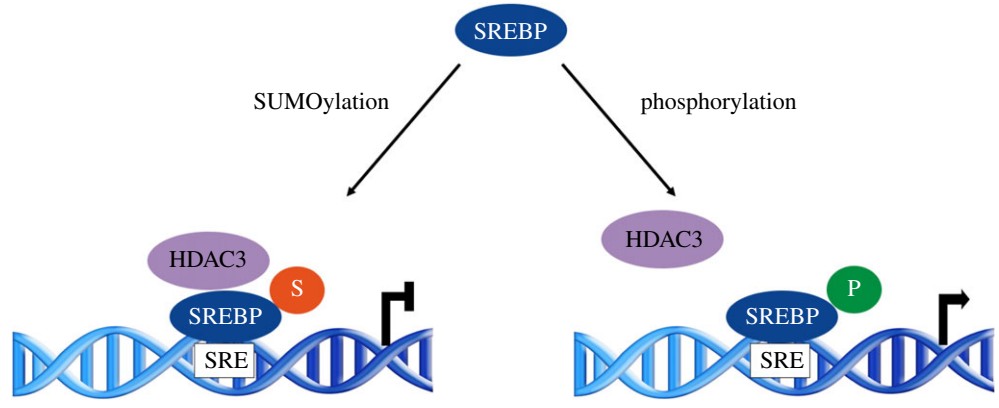

**Figure 2.** SUMOylation of SREBP2 reduces its transcriptional activity. SUMOylation of SREBP2 recruits HDAC3-containing complex and reduces its transcriptional activity. SREBP2 phosphorylation by MAPKs inhibits SUMOylation and activates the transcriptional activity of genes that contain sterol-response elements (SREs).

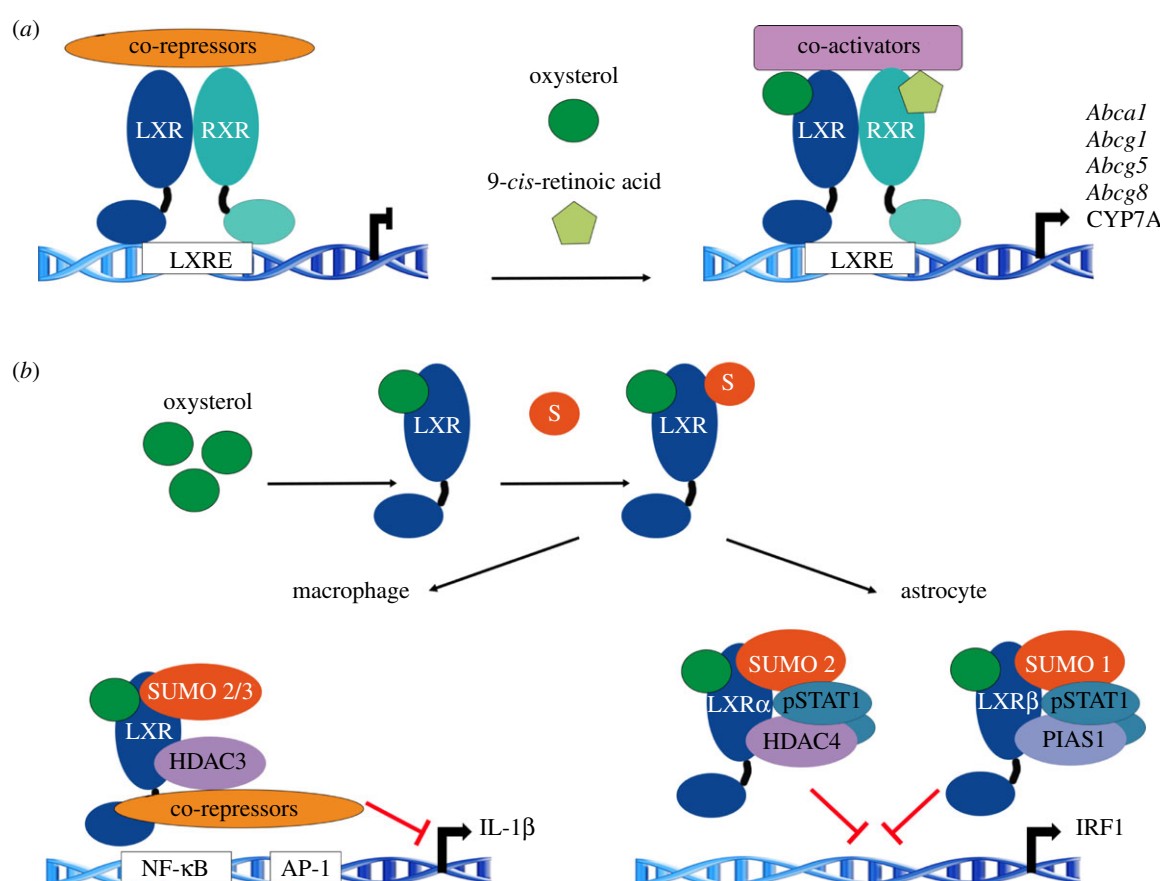

**Figure 3.** Intracellular sterols mediate SUMOylation of LXR. (*a*) LRX/RXR heterodimers bind to LXR response elements (LXREs) in promoters and repress the transcription of target genes by the recruitment of co-repressors such as NCoR1. Upon ligand binding, co-repressors are cleared and exchanged for co-activators leading to transcription. (*b*) Oxysterol-activated LXRs are conjugated to SUMO. SUMOylated LXR monomers bind to repressive complexes bound to promoters of pro-inflammatory target genes.

are oxygenated derivatives of cholesterol and intermediate precursors in the cholesterol biosynthesis pathway (such as desmosterol), constitute the LXRs endogenous ligands [69]. Oxysterols can also block the activation of SREBP-2. In the absence of activating ligand, the LXR–RXR heterodimer is bound to the response element on the DNA. In this basal state, the heterodimer complex recruits co-repressors such as nuclear co-repressor 1 (NCoR1) and a silencing mediator of retinoic acid and thyroid hormone receptor (SMRT) that maintains the chromatin in a repressive transcriptional state. The binding of a ligand induces conformational changes leading to the release of co-repressors and the recruitment of co-activators

(figure 3*a*). The LXR family consists of two isoforms: LXRα (encoded by *NR1H3*), expressed in liver, intestine, kidney and macrophages, and LXRβ (encoded by *NR1H2*), ubiquitously expressed [70]. These cholesterol-sensing transcription factors regulate genes involved in sterol uptake and transport in enterocytes [71,72]. In addition to preventing cholesterol accumulation in enterocytes, LXRs control the expression of genes involved in sterol secretion and catabolism in macrophages and liver (see §4).

LXR activation is modulated by deacetylation, phosphorylation [73] and SUMOylation [74]. Interestingly, oxysterols promote the SUMOylation of LXR and subsequent

transcriptional repression of pro-inflammatory genes [75,76] (figure 3b). In macrophages and hepatocytes, ligand-activated SUMOylation of LXR stabilizes co-repressors on NF-κB and downregulates the expression of target genes. In astrocytes, SUMOylation of LXRα and LXRβ prevents gene expression by blocking the binding of STAT1 to promoters. In LXRα, SUMO2 conjugation is mediated by the E3 ligase activity of HDAC4, whereas in LXRβ SUMO1 conjugation is mediated by the protein inhibitor of activated STAT1 (PIAS1) (figure 3b) [75]. Similar to LXR, ligand-induced SUMOylation of FXR, LRH-1 and peroxisome proliferator-activated receptor gamma (PPARγ) are as well involved in the repression of pro-inflammatory genes [74,76,77]. In invertebrates, SUMOylation restrains the systemic inflammation in *Drosophila* [78]. However, cholesterol-mediated SUMOylation in the control of inflammation has not been described. In addition, recent studies show the LXR-dependent repression of inflammatory genes despite the mutation of LXR SUMOylation sites [79,80]. These studies show that the mechanisms by which LXR protects from atherosclerosis are not totally understood. Thus, the relevance of LXR SUMOylation on sterol and inflammation requires further investigation.

# 4. SUMOylation in cholesterol transport and catabolism

The liver plays an important role in cholesterol metabolism by several pathways that include not only *de novo* cholesterol synthesis and dietary cholesterol uptake, but also reverse cholesterol transport (RCT), bile acid synthesis and biliary cholesterol excretion. Members of the nuclear receptor family such as LXR, LRH-1 and PPAR are involved in these pathways. In the following sections, we review what is published in the literature on the role of SUMO in these processes.

## 4.1. SUMOylation of LRH-1 in the RCT pathway

By RCT, the excess of cholesterol is removed from peripheral tissues, enters the circulation and is delivered to the liver for conversion into bile acids and excretion. RCT is essential to maintain cellular cholesterol homeostasis and is an important factor in atherosclerosis development. Cellular cholesterol efflux from macrophages is mediated through the action of ABCA1 and ABCG1 transporters. The main lipoprotein involved in the RCT is the high-density lipoprotein cholesterol (HDL-c). The HDL-c delivery to the liver can follow two routes: a direct one through binding to hepatic receptor scavenger receptor class B type 1 (SR-B1) or/and an indirect route via apoB-containing lipoproteins, VLDL (very low-density lipoprotein) or LDL. In addition to the selective uptake of cholesterol, SR-B1 function in the liver includes also biliary cholesterol secretion. Another pathway for the excretion of cholesterol is the *trans*-intestinal cholesterol efflux (TICE) via the small intestine.

Studies in mice have shown the biological impact of LRH-1 SUMOylation on RCT (figure 4). The nuclear receptor LRH-1 affects the expression of genes involved in cholesterol flux and RCT such as *Scavenger Receptor Class B Member 1* (*Scarb1*), *Abcg5* and *Abcg8* [81,82]. LRH-1 is modified by SUMO on several lysine residues, and the major effect of this modification is the repression of its transcriptional capacity [83,84]. SUMOylated LRH-1 recruits the co-repressor prospero homeobox protein 1 (PROX1) and therefore inhibits the LRH-1-dependent transcription of genes involved in RCT

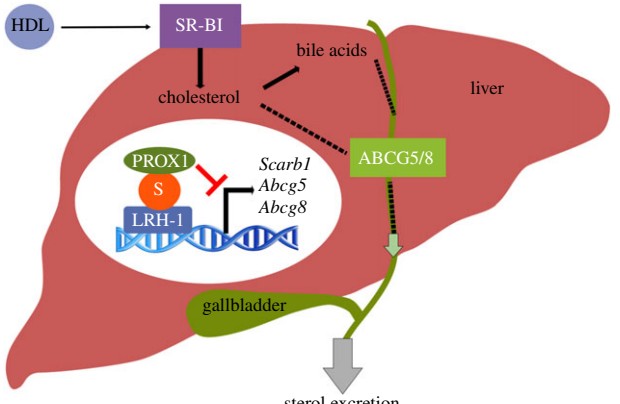

**Figure 4.** SUMOylation of LRH-1 inhibits RCT. SUMOylation of LRH-1 promotes interaction with the transcriptional repressor PROX1 and inhibits the LRH-1-dependent transcription of genes involved in hepatic RCT such as *Scarb1*, *Abcg5* and *Abcg8*. S: SUMO.

[85]. Accordingly, loss of LRH-1 SUMOylation by mutation on K289 leads to an increase in RCT. This study shows the *in vivo* function of LRH-1 SUMOylation in cholesterol homeostasis and atherosclerosis. Other studies support the notion that the mechanistic feature of LRH-1 SUMOylation is the promotion of protein–protein interactions. For example, SUMOylation of the human LRH-1 K224, the lysine residue corresponding to mouse LRH-1 K289, binds to a transcriptional co-repressor complex consisting of NCOR1, HDAC3 and G Protein Pathway Suppressor 2 (GPS2) in hepatoma cells [76]. The SUMO modification of LRH-1 is highly conserved in other organisms such as *C. elegans* and *D. melanogaster*, for which its function in sterol uptake in steroidogenic tissues has been described in §2 [52,86]. LXRs are also well-known regulators of RCT by inducing the expression of ABCA1, ABCG1 and ABCG5/G8 in macrophages and liver; however, the role of LXR SUMOylation in RCT is unknown.

## 4.2. FXR and SHP SUMOylation in cholesterol catabolism

Bile acid synthesis is an important route for cholesterol catabolism in the liver. Under conditions of low dietary cholesterol, the conversion of cholesterol into bile acids is reduced. The rate-limiting step in the classical bile acid biosynthetic route is the enzyme cholesterol 7a-hydroxylase (CYP7A1). The nuclear receptor FXR heterodimerizes with RXR and acts as an intracellular bile acid sensor that controls bile acid synthesis and transport [87]. FXR suppress the bile acid synthesis pathway through a feedback regulation involving LRH-1 and the orphan nuclear receptor Small Heterodimer Partner (SHP, NR0B2) (figure 5a) [88]. In the absence of bile acids, LRH-1, in concert with LXRα, stimulates the expression of the enzyme CYP7A1. In response to bile acids, FXR induces the expression of SHP, which in turn inhibits LRH-1, preventing the activation of bile acid synthesis. FXR is modified by SUMO1 in HepG2 cells on the conserved residues K122 and K275 in the activation function-1 (AF-1) and ligand-binding domains, respectively, which attenuates its capacity to function as a transcriptional activator [89]. SUMO1 decreased the ligand-dependent binding and/or the recruitment of FXR/RXRα to the *SHP* promoter [89]. Moreover, bile acid homeostasis is also maintained through SHP SUMOylation. When

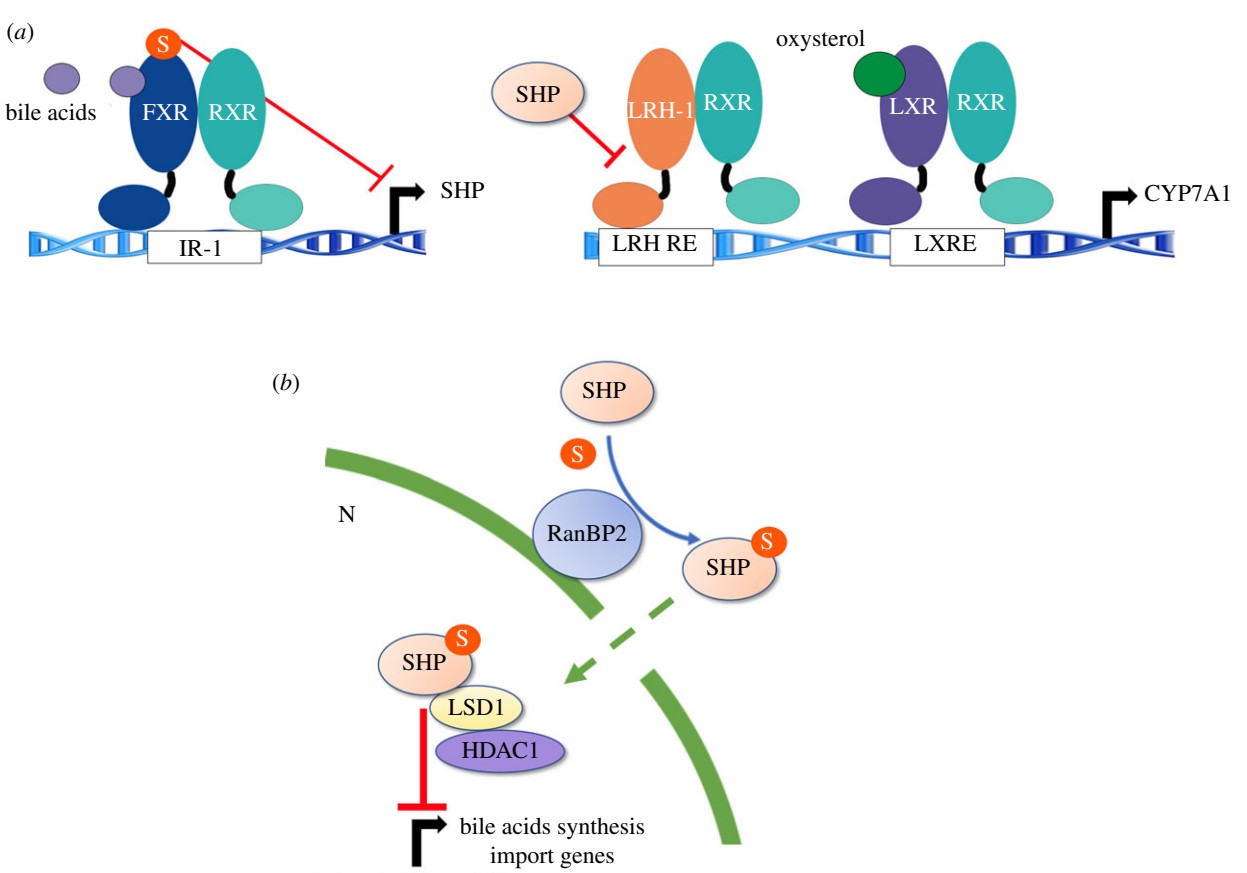

**Figure 5.** FXR and SHP SUMOylation in cholesterol catabolism. (*a*) Crosstalk between LXR and FXR–SHP–LRH-1 regulatory cascades in hepatic cholesterol catabolism. In the absence of bile acids, LRH-1 with LXR stimulates bile acid synthesis. The ligand SUMOylation of FXR attenuates its capacity to function as a transcriptional activator. (*b*) In response to elevated hepatic bile acids, SHP is modified by SUMO2-mediated by RanBP2. The modification facilitates nuclear transport and interaction with repressive histone-modifying enzymes, LSD1 and HDAC1, to inhibit bile acid synthetic genes.

the hepatic bile acid levels are elevated, the E3 ligase Ran-binding protein 2 (RanBP2/Nup358) mediates the SUMO2 modification of SHP. The SUMOylation of SHP facilitates its nuclear transport and increases its interaction with repressive histone-modifying enzymes LSD1 and HDAC1, which results in repression of bile acid synthesis and import transport (figure 5*b*) [90].

Recent studies in mice showed that FXR and its targets cholesterol exporter ABCG5/G8 stimulated the flux of cholesterol through the non-biliary TICE pathway in the intestinal lumen [91]. PPARδ activation also increased faecal neutral sterol excretion in mice, in part mediated by TICE [92]. However, the TICE route is less studied compared to RCT, and it is unknown whether SUMOylation of FXR, PPARs or other nuclear receptors modifies their recruitment to the promoters of genes involved in this pathway.

# 5. Concluding remarks

Research during the last decades has proved the critical role of SUMOylation in regulating almost all aspects of metabolism. Cholesterol is a very important cellular molecule that is involved in diseases such as atherosclerosis. SUMOylation modifies factors involved in cholesterol homeostasis that include SREBPs and members of the nuclear receptor superfamily such as LXR, FXR, LRH-1 and PPAR. These receptors, due to their function on cholesterol and bile acid homeostasis, are potential therapeutic targets. Indeed, the contribution of

SUMOylation in enhancing cholesterol efflux could lead to the development of treatments to control cardiovascular and other diseases. However, more studies are required to avoid the undesired hepatic lipogenesis. Another area with potential therapeutic applications could be the inhibition of inflammation by activating specific SUMO-dependent nuclear receptor transrepression pathways.

SUMO modification of nuclear receptors could also regulate their crosstalk. Therefore, it will be interesting to explore the implication of SUMO in heterodimer formation or the induced expression of other receptors in the context of cholesterol metabolism and associated diseases. As examples, RXR and PPAR, which are also negatively regulated by SUMO, form heterodimers with LXRs and FXRs in cholesterol metabolism [93]. The activation of the PPAR family in macrophages induces LXR gene expression and LXRα-dependent cholesterol efflux through ABCA1 [94]. Nevertheless, how SUMOylation specifically regulates the nuclear receptors and other transcription factors in cholesterol homeostasis, alone or in combination with other post-translational modifications, needs further investigation. In addition, to get insights on the E3 ligases and proteases involved in the SUMOylation of crucial factors in cholesterol metabolism regulation could open new venues for therapeutic intervention.

Data accessibility. This article has no additional data.

Authors' contributions. All authors contributed to the text and figures of this review.

Competing interests. We declare we have no competing interests.

**Acknowledgements.** We apologize to those whose related publications could not be cited due to space limitations. We are grateful to all members of Barrio's Lab for comments and suggestions. R.B. acknowledges grant nos. BFU2017-84653-P (MINECO/AEI/FEDER/EU), SEV-2016-0644 (Severo Ochoa Excellence Program, MINECO/AEI), 765445-EU (UbiCODE Program, EU) and SAF2017-90900-REDT (UBIRed Program, MINECO/AEI).

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
