## [Reviewer comments · Open Biology]

Review History

RSOB-20-0054.R0 (Original submission)

Review form: Reviewer 1

Recommendation

Accept with minor revision (please list in comments)

Do you have any ethical concerns with this paper?

No

Comments to the Author

In this manuscript, the authors review the literature concerning the role of SUMOylation in the regulation of cholesterol metabolism. They begin by a detailed introduction on SUMOylation, in particular the enzymatic cascade responsible for its conjugation/deconjugation. It is followed by a presentation of the main actors of cholesterol metabolism. Then, they describe the connections between SUMOylation and cholesterol metabolism, in particular how it regulates the expression of critical actors of this pathway. The manuscript is clear, well written and the figures a nicely illustrating the different parts.

I however suggest some modifications. My main point concerns the introduction on SUMOylation. It would be interesting to add, after the part 1.2, a part on the roles of

SUMOylation. In particular, it should emphasize the role of SUMOylation in the control of gene expression as this is the main link between SUMOylation and cholesterol metabolism.

The molecular consequences of SUMOylation, in particular the ability of SUMOylated proteins to recruit SIM-containing partners should also be explained and exemplified in the context of gene expression.

The part on the SUMO conjugation/deconjugation pathway could be shortened and the reader referred to reviews describing it in details.

Review form: Reviewer 2

Recommendation

Accept with minor revision (please list in comments)

Do you have any ethical concerns with this paper?

No

Comments to the Author

The authors have written a review linking SUMOylation as a regulatory mechanism for nuclear receptor transcription factors and enzymatic protein activity with cholesterol metabolism. It is a relevant review since most information in the field is disperse, and alterations of cholesterol levels and cholesterol homeostasis are associated to developmental defects and relevant health issues. The Figures of the article are illustrative and help the comprehension of the text.

There are several minor issues that deserve some attention:

- 1) Figure 1 is too small. A larger figure is recommended. Also this Figure is referred only once in the text. Since the point 1.2 explains the mechanistics of SUMO conjugation and deconjugation, one or two references more in the text to Figure 1 will help the general reader.
- 2) Sentence in lines 145 to 149 explains that several nuclear receptor transcription factors regulate bile acids synthesis and cholesterol catabolism in mammals. However, no target genes are provided. A short sentence with further explanation on how this regulation impacts on cholesterol levels/metabolism will help the readers to understand the relevance of this regulatory action.
- 3) Subsection 2.3 is dedicated to the regulation of SREBP-2 activity by SUMOylation. The previous subsections 2.1 and 2.2 compare the knowledge gathered in invertebrate animal models to that gathered in vertebrate/mammals. However, subsection only refers to mammals. To be consistent with this previous organization, is there any relevant piece of information concerning SUMO and SREBP in invertebrates? If there is not, authors should include a short statement on this.

The same comment applies to section 3.

- 4) I think that there is an en dash missing in the title of Section 3 (also in some parts of the text) "Cholesterol-mediated LXR sumoylation..."
- 5) Sentence on lines 245- 247, the sentence is not clear. Is sumoylated LXR stabilizing co-repressors on DNA target motifs, or on DNA boxes recognized by other co-repressor complexes? A longer sentence may be help the reader to understand the point.
- 6) The following sentence, lines 247-250 (referring to Figure 3B) will be also clearer if rephrased, since the second part of the sentence has no verbs.
- 7) Line 252, the term "regulation" is unspecific in this sentence. I recommend substituting it by a more specific term, e.g. trans-activation, downregulation, repression. Same comment applies to line 295 and the term "targeting". Again, meaning would be clearer if substituted by an specific action, such as trans-activation, downregulation...
- 8) General comment.

Very few E3 SUMO ligases and proteases are mentioned concerning SUMO1 and SUMO2 modification of proteins related to cholesterol metabolism. Also, why and how these SUMO pathway enzymes are regulated by cholesterol levels or metabolites is not explained. Most probably these enzymes have not been identified for most cases, but if there is some knowledge, it would be advisable to introduce some sentences in the text since they may be potential targets for therapeutic intervention.

Decision letter (RSOB-20-0054.R0)

20-Mar-2020

Dear Dr Barrio,

We are pleased to inform you that your manuscript RSOB-20-0054 entitled "SUMOylation in the control of Cholesterol Homeostasis" has been accepted by the Editor for publication in *Open Biology*. The reviewer(s) have recommended publication, but also suggest some minor revisions to your manuscript. Therefore, we invite you to respond to the reviewer(s)' comments and revise your manuscript.

Please submit the revised version of your manuscript within 7 days. If you do not think you will be able to meet this date please let us know immediately and we can extend this deadline for you.

- 1) A text file of the manuscript (doc, txt, rtf or tex), including the references, tables (including captions) and figure captions. Please remove any tracked changes from the text before submission. PDF files are not an accepted format for the "Main Document".
- 2) A separate electronic file of each figure (tiff, EPS or print-quality PDF preferred). The format should be produced directly from original creation package, or original software format. Please note that PowerPoint files are not accepted.
- 3) Electronic supplementary material: this should be contained in a separate file from the main text and meet our ESM criteria (see <http://royalsocietypublishing.org/instructions-authors#question5>). All supplementary materials accompanying an accepted article will be treated as in their final form. They will be published alongside the paper on the journal website and posted on the online figshare repository. Files on figshare will be made available

approximately one week before the accompanying article so that the supplementary material can be attributed a unique DOI.

Online supplementary material will also carry the title and description provided during submission, so please ensure these are accurate and informative. Note that the Royal Society will not edit or typeset supplementary material and it will be hosted as provided. Please ensure that the supplementary material includes the paper details (authors, title, journal name, article DOI). Your article DOI will be 10.1098/rsob.2016[last 4 digits of e.g. 10.1098/rsob.20160049].

4) A media summary: a short non-technical summary (up to 100 words) of the key findings/importance of your manuscript. Please try to write in simple English, avoid jargon, explain the importance of the topic, outline the main implications and describe why this topic is newsworthy.

Images

Data-Sharing

It is a condition of publication that data supporting your paper are made available. Data should be made available either in the electronic supplementary material or through an appropriate repository. Details of how to access data should be included in your paper. Please see <http://royalsocietypublishing.org/site/authors/policy.xhtml#question6> for more details.

Data accessibility section

Sincerely,

The Open Biology Team

<mailto:openbiology@royalsociety.org>

Reviewer(s)' Comments to Author:

Referee: 1

Comments to the Author(s)

In this manuscript, the authors review the literature concerning the role of SUMOylation in the regulation of cholesterol metabolism. They begin by a detailed introduction on SUMOylation, in particular the enzymatic cascade responsible for its conjugation/deconjugation. It is followed by a presentation of the main actors of cholesterol metabolism. Then, they describe the connections between SUMOylation and cholesterol metabolism, in particular how it regulates the expression of critical actors of this pathway. The manuscript is clear, well written and the figures a nicely illustrating the different parts.

I however suggest some modifications. My main point concerns the introduction on SUMOylation. It would be interesting to add, after the part 1.2, a part on the roles of

SUMOylation. In particular, it should emphasize the role of SUMOylation in the control of gene expression as this is the main link between SUMOylation and cholesterol metabolism.

The molecular consequences of SUMOylation, in particular the ability of SUMOylated proteins to recruit SIM-containing partners should also be explained and exemplified in the context of gene expression.

The part on the SUMO conjugation/deconjugation pathway could be shortened and the reader referred to reviews describing it in details.

Referee: 2

Comments to the Author(s)

The authors have written a review linking SUMOylation as a regulatory mechanism for nuclear receptor transcription factors and enzymatic protein activity with cholesterol metabolism. It is a relevant review since most information in the field is disperse, and alterations of cholesterol levels and cholesterol homeostasis are associated to developmental defects and relevant health issues. The Figures of the article are illustrative and help the comprehension of the text.

There are several minor issues that deserve some attention:

1) Figure 1 is too small. A larger figure is recommended. Also this Figure is referred only once in the text. Since the point 1.2 explains the mechanistics of SUMO conjugation and deconjugation, one or two references more in the text to Figure 1 will help the general reader.

2) Sentence in lines 145 to 149 explains that several nuclear receptor transcription factors regulate bile acids synthesis and cholesterol catabolism in mammals. However, no target genes are provided. A short sentence with further explanation on how this regulation impacts on cholesterol levels/metabolism will help the readers to understand the relevance of this regulatory action.

3) Subsection 2.3 is dedicated to the regulation of SREBP-2 activity by SUMOylation. The previous subsections 2.1 and 2.2 compare the knowledge gathered in invertebrate animal models to that gathered in vertebrate/mammals. However, subsection only refers to mammals. To be consistent with this previous organization, is there any relevant piece of information concerning SUMO and SREBP in invertebrates? If there is not, authors should include a short statement on this.

The same comment applies to section 3.

4) I think that there is an en dash missing in the title of Section 3 (also in some parts of the text) "Cholesterol-mediated LXR sumoylation..."

5) Sentence on lines 245- 247, the sentence is not clear. Is sumoylated LXR stabilizing co-repressors on DNA target motifs, or on DNA boxes recognized by other co-repressor complexes? A longer sentence may be help the reader to understand the point.

6) The following sentence, lines 247-250 (referring to Figure 3B) will be also clearer if rephrased, since the second part of the sentence has no verbs.

7) Line 252, the term "regulation" is unspecific in this sentence. I recommend substituting it by a more specific term, e.g. trans-activation, downregulation, repression.

Same comment applies to line 295 and the term "targeting". Again, meaning would be clearer if substituted by an specific action, such as trans-activation, downregulation...

8) General comment.

Very few E3 SUMO ligases and proteases are mentioned concerning SUMO1 and SUMO2 modification of proteins related to cholesterol metabolism. Also, why and how these SUMO pathway enzymes are regulated by cholesterol levels or metabolites is not explained. Most probably these enzymes have not been identified for most cases, but if there is some knowledge, it would be advisable to introduce some sentences in the text since they may be potential targets for therapeutic intervention.

Author's Response to Decision Letter for (RSOB-20-0054.R0)

See Appendix A.

Decision letter (RSOB-20-0054.R1)

14-Apr-2020

Dear Dr Barrio

We are pleased to inform you that your manuscript entitled "SUMOylation in the control of Cholesterol Homeostasis" has been accepted by the Editor for publication in Open Biology.

You can expect to receive a proof of your article from our Production office in due course, please check your spam filter if you do not receive it within the next 7-10 working days. Please let us know if you are likely to be away from e-mail contact during this time.

Sincerely,
The Open Biology Team
mailto:openbiology@royalsociety.org

Appendix A

Reviewer(s)' Comments to Author:

Referee: 1

Comments to the Author(s)

In this manuscript, the authors review the literature concerning the role of SUMOylation in the regulation of cholesterol metabolism. They begin by a detailed introduction on SUMOylation, in particular the enzymatic cascade responsible for its conjugation/deconjugation. It is followed by a presentation of the main actors of cholesterol metabolism. Then, they describe the connections between SUMOylation and cholesterol metabolism, in particular how it regulates the expression of critical actors of this pathway. The manuscript is clear, well written and the figures a nicely illustrating the different parts.

I however suggest some modifications. My main point concerns the introduction on SUMOylation. It would be interesting to add, after the part 1.2, a part on the roles of SUMOylation. In particular, it should emphasize the role of SUMOylation in the control of gene expression as this is the main link between SUMOylation and cholesterol metabolism.

As suggested by the Reviewer, we have included a new section (1.3) in the introduction related to the role of SUMO in the control of gene expression discussing the regulation of transcription factors and histone modification and chromatin remodeling.

The molecular consequences of SUMOylation, in particular the ability of SUMOylated proteins to recruit SIM-containing partners should also be explained and exemplified in the context of gene expression.

We have included the following paragraph to explain the ability of SUMOylated proteins to recruit SIM containing partners with examples:

“In addition to the covalent attachment of SUMO to lysine residues in target proteins, the SUMO-interacting motifs (SIMs) mediate non-covalent interactions with SUMO or SUMO-conjugated proteins (reviewed in Kerscher 2007). The functional consequences of SUMO binding to SIM-containing proteins include the recruitment of proteins to subcellular localizations, the formation of promyelocytic leukaemia protein (PML) nuclear bodies or the recruitment of repressors complexes to the chromatin {Shen, 2006;Lin, 2006}.

The part on the SUMO conjugation/deconjugation pathway could be shortened and the reader referred to reviews describing it in details.

We have shortened the part on the SUMO conjugation/deconjugation by eliminating the last sentence: “Two additional SUMO proteases have been described in humans, deSUMOylating isopeptidase (DeSI), and the ubiquitin-specific protease-like 1 (USPL1)” to preserve the general meaning.

Referee: 2

Comments to the Author(s)

The authors have written a review linking SUMOylation as a regulatory mechanism for nuclear receptor transcription factors and enzymatic protein activity with cholesterol metabolism. It is a relevant review since most information in the field is dispersed, and alterations of cholesterol levels and cholesterol homeostasis are associated to developmental defects and relevant health issues. The Figures of the article are illustrative and help the comprehension of the text.

There are several minor issues that deserve some attention:

1) Figure 1 is too small. A larger figure is recommended. Also this Figure is referred only once in the text. Since the point 1.2 explains the mechanistics of SUMO conjugation and deconjugation, one or two references more in the text to Figure 1 will help the general reader.

We have replaced Figure 1 with a larger one and have included two additional references in the text to Figure 1.

2) Sentence in lines 145 to 149 explains that several nuclear receptor transcription factors regulate bile acids synthesis and cholesterol catabolism in mammals. However, no target genes are provided. A short sentence with further explanation on how this regulation impacts on cholesterol levels/metabolism will help the readers to understand the relevance of this regulatory action.

We have added a paragraph about the role of the nuclear receptors in the regulation of bile acid synthesis and excretion. We have included as example a target gene of two nuclear receptors with leads to different impact on cholesterol metabolism:

“The levels of bile acids are tightly regulated through the transcriptional regulation of the cytochrome P450 7A1 (CYP7A1), rate limiting enzyme in the pathway of bile acid biosynthesis (Russell and Setchell 1992). As examples, LXR α upregulates CYP7A1 mRNA and eliminates the excess of cholesterol via bile acid synthesis and excretion (Peet et al., 1998), whereas FXR acts as a receptor for bile acids and inhibits bile acid synthesis by repressing CYP7A1 transcription (Makishima et al., 1999).”

3) Subsection 2.3 is dedicated to the regulation of SREBP-2 activity by SUMOylation. The previous subsections 2.1 and 2.2 compare the knowledge gathered in invertebrate animal models to that gathered in vertebrate/mammals. However, subsection only refers to mammals. To be consistent with this previous organization, is there any relevant piece of information concerning SUMO and SREBP in invertebrates? If there is not, authors should include a short statement on this.

We have included a short statement about the SREBP in invertebrates. The only SREBP member is not involved in cholesterol metabolism and no information about their SUMOylation is available:

“In invertebrates such as *C. elegans* and *Drosophila* there is only one SREBP, which resembles the vertebrate SREBP1c and controls fatty acid biosynthesis (Theopold et al., 1996; Horton et al., 2002). In these organisms, the *Drosophila* dSREBP and the *C. elegans* SBP-1 do not seem to be involved in cholesterol homeostasis; their activity are regulated by phospholipids, but not by sterols (Dobrossotskaya et al., 2002; Seegmiller et al., 2002).”

The same comment applies to section 3.

For Section 3, there is no clear evidence for cholesterol mediated SUMOylation in the control of inflammation. We added two sentences in this section to explain this:

“In invertebrates, SUMOylation restrain the systemic inflammation in *Drosophila* (Paddibhatla et al., 2010). However, cholesterol mediated SUMOylation in this control of inflammation has not been described.”

4) I think that there is an en dash missing in the title of Section 3 (also in some parts of the text) "Cholesterol-mediated LXR sumoylation..."

We have added an en dash in the title and in others parts when the text required it.

5) Sentence on lines 245- 247, the sentence is not clear. Is sumoylated LXR stabilizing co-repressors on DNA target motifs, or on DNA boxes recognized by other co-repressor complexes? A longer sentence may be help the reader to understand the point.

The sentence on lines 245-247 does not relate to SUMOylation. It refers to the mechanism of nuclear receptors shown in Figure 3A. The SUMOylation is explained in the next sentences and refers to Figure 3B. We have explained this better in two sentences:

“In the absence of activating ligand, the LXR-RXR heterodimer is bound to the response element on the DNA. In this basal state, the heterodimer complex recruits co-repressors such as nuclear co-repressor 1 (NCoR1) and silencing mediator of retinoic acid and thyroid hormone receptor (SMRT) that maintain the chromatin in a repressive transcriptional state. The binding of a ligand, induces conformational changes leading to the release of co-repressors and recruitment of co-activators (Figure 3A).”

6) The following sentence, lines 247-250 (referring to Figure 3B) will be also clearer if rephrased, since the second part of the sentence has no verbs.

To make the sentence in lines 247-250 clearer we have rephrased it in two sentences:

“In astrocytes, SUMOylation of LXR α and LXR β prevents gene expression by blocking the binding of STAT1 to promoters. In LXR α , SUMO2 conjugation is mediated by the E3 ligase activity of HDAC4, whereas in LXR β SUMO1 conjugation is mediated by protein inhibitor of activated STAT1 (PIAS1) (Figure 3B)”

7) Line 252, the term "regulation" is unspecific in this sentence. I recommend substituting it by a more specific term, e.g. trans-activation, downregulation, repression. Same comment applies to line 295 and the term "targeting". Again, meaning would be clearer if substituted by an specific action, such as trans-activation, downregulation...

As suggested by the Reviewer, we have changed the term "regulation" by "repression" (line 293) and the term "targeting" by "inducing the expression of" (line 336).

8) General comment.

Very few E3 SUMO ligases and proteases are mentioned concerning SUMO1 and SUMO2 modification of proteins related to cholesterol metabolism. Also, why and how these SUMO pathway enzymes are regulated by cholesterol levels or metabolites is not explained. Most probably these enzymes have not been identified for most cases, but if there is some knowledge, it would be advisable to introduce some sentences in the text since they may be potential targets for therapeutic intervention.

We agree with the Reviewer that the identification of ligases and proteases in the mentioned pathways will be very interesting as potential targets for therapeutic intervention. However, we could not find much information on how those few ligases and proteases are regulated by cholesterol. Therefore, we just added a sentence stating the fact that it would be important to get knowledge in this aspect of cholesterol regulation.